# Role of the Molecular Mass on the Elastic Properties of Hybrid Carrageenan Hydrogels

**DOI:** 10.3390/gels11010077

**Published:** 2025-01-20

**Authors:** Gabriela Gonçalves, Bruno Faria, Izabel Cristina Freitas Moraes, Loic Hilliou

**Affiliations:** 1Institute for Polymers and Composites, University of Minho, 5800-048 Guimarães, Portugal; pg50598@alunos.uminho.pt (G.G.); bruno.faria@dep.uminho.pt (B.F.); 2Department of Food Engineering, Faculty of Animal Science and Food Engineering (FZEA), University of São Paulo (USP), Pirassununga 13635-900, SP, Brazil; bel@usp.br

**Keywords:** carrageenan, hydrogel, molecular mass, shear storage modulus

## Abstract

A set of carrageenans produced in the potassium form and with chemical structures varying from pure iota-carrageenans to nearly pure kappa-carrageenans is submitted to ultrasonication to reduce their molecular masses Mw while maintaining a constant chemical structure and a polydispersity index around 2. The kinetics of ultrasound-induced chain scission are found to be slower for polysaccharides richer in kappa-carrageenan disaccharide units. From the elasticity of samples directly gelled in a rheometer at 1 *w*/*v*% in 0.1 M potassium chloride, a critical molecular mass Mc is identified as the mass below which no gel can be formed. Mc is found to be smaller for kappa- and kappa-2-carrageenans of the order of 0.13–0.21 MDa. The presence of more sulphated disaccharide units significantly increases Mc up to 0.28 MDa for iota-carrageenan and 0.57 MDa for a highly sulphated hybrid carrageenan. For the set of Mw and carrageenans tested, no plateau in the Mw dependence of the gels’ elasticities is found.

## 1. Introduction

One of the most prominent characteristics of carrageenans, a family of sulphated polysaccharides contained in red seaweeds, is perhaps their diversity [1]. This diversity emanates essentially from variations in their chemical structure, originating from the algae they are sourced from and the extraction route used to produce a food additive [2] or an ingredient for the pharmaceutical industry [3]. These polysaccharides are traditionally divided into eight types, according to the number and position of the sulphate groups in the repeating units. The linear polyelectrolyte chain of carrageenans is formed by repeating disaccharide units (diads) of β-D-galactopyranose (labelled as **G**) and α-D-galactopyranose (**D**) or 3,6-anhydrogalactose (**DA**). The diads are linked together by carbons 3 of **G** and 4 of **D** (or **DA**). The letter codes assigned to the residues correspond to the simplified denomination developed by Knutsen et al. [4]. Sulphated galactans are classified based on the presence of the 3,6-anhydro-bridge on the 4-linked galactose residue, as well as the position and number of sulphate groups. For example, the iota-carrageenan diad is made of β-D-galactopyranose sulphated at the fourth carbon (**G4S**) linked to 3,6-anhydrogalactose-D-galactopyranose sulphated at the second carbon (**DA2S**). Thus, iota-carrageenan diad is annotated as **G4S-DA2S**, whereas the kappa-carrageenan diad is **G4S-DA**, and their respective biological precursors, nu and mu, are noted as **G4S-D2S,6S** and **G4S-D6S**, respectively. The **G4S** group is common to all these diads and, therefore, characterizes this family of kappa-carrageenans (see [5] for an illustration of the chemical structures of diads from this family).

There is an additional degree of variety and complexity when moving from the diad to the polysaccharide. The most commercially used carrageenans are indeed made of diads from the kappa-carrageenan family with kappa-carrageenan forming strong gels and iota-carrageenan forming weaker gels. Kappa-carrageenan is essentially made from roughly 90 mol.% of **G4S-DA** [6], whereas iota-carrageenan shows an even more homopolymer-like structure with 90 to 96 mol.% **G4S-DA2S** [7,8]. Over the past 20 years, another industrially relevant class has emerged, hybrid carrageenans, also known as kappa-2 or weak kappa-carrageenan [9]. Kappa-2-carrageenan has a more heteropolymer structure, which is better described as a random block copolymer, where the blocks are sequences of diads from the kappa-carrageenan family [5].

The thermal and mechanical properties of gels made from kappa-carrageenan, iota-carrageenan and their blends have extensively been studied, see, e.g., [1,10,11]. During the cooling of a hot carrageenan solution in the presence of cations such as K^+^, Na^+^ or Ca^2+^, helical conformers self-aggregate into a three-dimensional network, enabling the transmission of mechanical stresses. The exact structure–elastic relationships within the network are still an unsettled topic [12] and many models have been advanced to distinguish between the mechanical properties of kappa- and iota-carrageenan gels as well as their blends. Interestingly, the role of the polysaccharide chain length on the gel properties has received much less attention. This, however, is an attractive topic given the more recent controversy on carrageenan toxicity [13], current regulations that limit the amount of low molecular mass fraction in food or pharmaceutical applications, and the possible leaching of this fraction from a carrageenan gel.

Three decades ago, Rochas et al. [14] published a seminal work on the existence of two critical molecular masses in kappa-carrageenan gelled in the presence of KCl. A set of carrageenan with different molecular masses Mw ranging from 0.006 MDa to 1.2 MDa (weight average) was prepared by ultrasonication or hydrolysis of a commercial kappa-carrageenan. Gel samples were then prepared at different polysaccharide concentrations and ionic strengths and submitted to compression testing. A first critical macromolecular mass Mmax was identified, signaling the limit above which the gel’s Young modulus becomes independent of Mw. Mmax was found to be 0.18 MDa, irrespective of ionic strength or carrageenan concentration. From the decrease in the Young modulus with the decrease in Mw below Mmax, a second critical mass Mc of 0.04 MDa was extrapolated that signals the minimum chain length needed to form a kappa-carrageenan gel when adding 10 g of polysaccharide in 1 L of 0.1 M KCl. At the end of the paper, the authors quote that “it is obvious that more information is required to understand the meaning of the value of Mmax and its possible relation to the mesh size of the gel”. The interplay between Mw and the gel elasticity has not been the focus of the recent literature, as most studies simultaneously vary both Mw and the chemical structure of the carrageenan [5]. An effort towards clearing up such interplay was made by comparing the gel elasticity of two polysaccharides having virtually identical compositions in carrageenan diads but with nearly an order of magnitude difference in Mw [15]. In NaCl, the hybrid carrageenan with the smaller Mw formed a gel with larger elasticity.

Here, a set of carrageenans with different chemical structures ranging from pure iota-carrageenan to pure kappa-carrageenan are prepared in the potassium form. Different Mw are produced by ultrasonication, and the gel properties of the resulting carrageenans are studied in the presence of KCl. The results, obtained in the presence of a single cation, are presented with a view to filling the gap of knowledge on the role of Mw on the carrageenan gel elasticity. Ultrasonication was chosen over hydrolysis because it is considered a nonrandom process, allowing the PDI to converge to a minimum value while preserving the chemistry [16,17,18,19]. Although the exact mechanism of bond scission by ultrasonication remains a matter of debate [17,19], significant chemical modifications are less likely to occur. The practical applicability of the results reported here lies in the fact that the critical mass Mc needed to form carrageenan gels depends on the carrageenan chemical structure. Below Mc, the corresponding carrageenan cannot be used as an edible gelling agent in the food or pharmaceutical industry [1,2].

## 2. Results and Discussion

### 2.1. Chemical Structure of Extracted Hybrid Carrageenans

The chemical structures of the five extracted carrageenans (ultrasonication time *t* = 0), determined by the analysis of the proton NMR spectra displayed in Figure 1, are reported in Table 1. Vertical dotted lines in Figure 1 indicate the peaks of the assigned disaccharide units showing up at chemical shifts referenced in [20]. As expected, extracts from *Spinosum* and *Cottonii* essentially resemble the chemical compositions of commercial iota- and kappa-carrageenans, respectively [6,7,8]. The extract from *Mastocarpus stellatus* can be classified as a kappa-2-carrageenan as it contains solely **G4S-DA** and **G4S-DA2S** diads, with the latter amounting 31 mol.%, which is well within the range defined by the industry for kappa-2-carrageenan [9]. The carrageenan from *Gigartina pistillata* is essentially a kappa/iota-hybrid carrageenan but with a significant amount of **G4S-D2S,6S** (see the corresponding peak indicated by a red arrow in Figure 1), which interrupts the formation of the helical conformation of blocks of **G4S-DA** and **G4S-DA2S [8]**. This will also be the case for the extract from *Gigartinale*, which shows a more heterogeneous structure as it contains all diads from the kappa-carrageenan family.

Overall, this set of carrageenans allows for questioning whether Mmax (and Mc) are universal values for carrageenans or if they depend on the detail of their chemical structures. The *Cottonii* extract provides an opportunity to reproduce results from the literature [14], whereas the *Spinosum* and *M. stellatus* extracts will be used to evaluate Mmax and Mc for iota-carrageenans and kappa-2-carrageenans, respectively. The impact of more sulphated diads, in a smaller amount but of different types and in a large amount of the same type, on the molecular mass dependence of the gelling properties will be studied with the remaining two extracts.

### 2.2. Kinetics of the Mw Degradation by Ultrasonication

The evolution of the molecular mass distribution of carrageenan chains as a function of the time of ultrasonication *t* is presented in Figure 2. The time scale of ultrasonication for iota-carrageenan does not allow for a clear assessment of the kinetics, as much of the chain scission occurred during the first hour, in harmony with the results recently reported for a kappa-carrageenan sample processed at 40 °C using an ultrasonic probe [17]. However, the data in Figure 2 clearly indicate that iota-carrageenan is much more sensitive to ultrasonication than the others, as Mw values saturate to nearly 0.25 MDa after three hours. In contrast to this, Mw is still decreasing after 5 h for the hybrid carrageenan from *G. pistillata*, whereas the kappa-carrageenan shows a saturation in Mw only after 16 h. Overall, the semi-logarithmic plot of Mw indicates that the break-up of carrageenan chains does not clearly follow the exponential kinetics reported elsewhere for porphyran, another algal sulphated polysaccharide [16]. As such, most of the data in Figure 2 cannot be fitted by a first-order kinetics equation on the basis of equal probability of scission of the α(1→3) and β(1→4) glycosidic linkages [16]. However, an exponential fit to the dataset for kappa-carrageenan was attempted up to 16 h and returned a rate constant of degradation reaction *k* = 20 ± 5 h^−1^, which is 60 times larger than the *k* reported for porphyran ultrasonicated at 60 °C with much higher power (1200 W against 400 W used here) [16]. Similar exponential fits to the whole datasets returned *k* values with large errors for the four remaining carrageenans, namely *k* = 10 ± 1 h^−1^ for iota-carrageenan, *k* = 7 ± 1 h^−1^ for *G. pistillata*, *k* = 43 ± 14 h^−1^ for *M. stellatus* and *k* = 21 ± 5 h^−1^ for the Gigartinale. Such bad fit quality confirms that the degradation kinetics do not clearly follow a first-order equation.

It has been shown that the rate constant *k* depends on the initial Mw of polystyrenes [18], with larger chains taking more time to degrade. Thus, differences in Figure 2 may not only relate to different carrageenan chemical structures but simply to the differences in the initial Mw of the polysaccharides. Given the width of the molecular mass distributions quantified by the polydispersity index, PDI, computed as the ratio Mw/Mn, where Mn is the number average of the molecular mass, the initial Mw of the iota-carrageenan (0.5 MDa) is comparable to the initial Mw of carrageenans from *M. stellatus* (0.65 MDa) and from the Gigartinale (0.66 MDa), which both show much slower degradation kinetics in spite of the two outlier data points at 16 h and 14 h, respectively. Indeed, these two anomalies may result from the temporary reaggregation of carrageenan chains during sample preparation for Mw determination by size exclusion chromatography. Alternatively, since these two carrageenans originated from separate flasks submitted to a noncontinuous sonication process due to the long hours required, this interruption may have favored chain reaggregation during rest periods, potentially affecting the sample viscosity and the subsequent ultrasonication [17,18,19]. Nevertheless, the chemical structure of the carrageenans seems to impact on their resistance to degradation by ultrasounds since the iota-carrageenan sample shows much faster kinetics in Figure 2. The same conclusion cannot be driven from the time dependence of the PDI, which converges to 2 after 6 h of ultrasonication for all carrageenans but the iota-carrageenan. The latter was liquid-like after sonicating during 2 h and, thus, was not processed for much longer *t*, as the main objective of the study is to identify a Mc for all carrageenans. The lack of conclusive results also comes from the initial PDI, which is much larger for the iota-carrageenan sample. To date, there is no consensus on the mechanism for explaining the ultrasound-assisted degradation of biopolymers [19]. However, it is taken as a nonrandom process where preferentially larger chains are first chopped into smaller chains and the break-up mostly occurs at the center of the macromolecules. Thus, the fact that the PDI drops to 2 for all carrageenans studied here comes as no surprise, in agreement with the limiting PDI found for kappa-carrageenans [17] and polystyrene [18], although the existence of a limiting PDI at a longer ultrasonication time is still a matter of debate for many biopolymers [19].

### 2.3. Gelling Behavior of Sonicated Carrageenans and Identification of Mc

The gelling behavior of the sonicated hybrid carrageenans from the Gigartinale is displayed in Figure 3a. Qualitatively similar cooling curves were measured with the other gelling carrageenans. The thermal dependence of the rheometer’s gap measured during cooling under controlled normal force (set to 0 N) gives a better estimate for the temperature of onset of gel setting *Ton* than the gelling temperature usually defined by the crossover between the shear elastic modulus G′ and the shear loss modulus G″ [21,22].

Indeed, the inset of Figure 3a shows that both G′ and G″ are noisy due to the small strain amplitude applied during cooling to avoid the shear-induced rearrangement of the network developing between the shearing plates. The mechanical spectra of the gels resulting from cooling are shown in Figure 3b, together with the spectrum recorded for the carrageenan sonicated during 16 h. The latter indicates a liquid-like viscoelastic behavior, since G″ (solid symbols) is larger than G′ (empty symbols) over the entire frequency range. The hybrid carrageenans submitted to shorter ultrasonication times display the mechanical spectra of gels, with G′ being virtually frequency-independent and much larger than G″. Note here that these spectra are more solid-like than the mechanical spectra of kappa- and iota-carrageenan gelled in a fermentation medium producing chain degradation [23]. Moreover, with a vertical shift in the curves, the mechanical spectra can be superimposed onto a master curve (result not shown), which suggests that the network structure is overall unchanged, even for 4 times shorter chains. The quality of the master curve is, however, lower for G″, as the local minimum shifts to lower frequencies for the larger chain (*t* = 0). This indicates that the slowest relaxation process at lower frequencies, not resolved in the spectra, is drifted to shorter times for the sonicated chains (the minimum moves to higher frequencies in Figure 3b), as expected from the faster dynamics of shorter chains.

The values of *G0* inferred from the mechanical spectra are listed for all carrageenans in Table 2, along with the temperatures for onset of gel setting *Ton*. Essentially, *Ton* decreases with the ultrasonication time, though an outlier identified for the Gigartinale at 14 h sonication also shows up in Table 2. Note here that only 0.0373 g of KCl was required to reach 0.1 M with this sample due to the small amount of sonicated carrageenan available. Thus, there may have been inaccuracies in weighing, with possible negative impact on the sample’s rheological properties.

The iota-carrageenan and the highly sulphated hybrid carrageenan from *G. pistillata* needed only a couple of hours of sonication to lose their gelling ability. This again suggests that the initial Mw is not a key parameter to assess the resistance of carrageenans to ultra-sounds (see also Figure 2a). There is a clear correlation between the sonication time *t* and *G0*, suggesting that the gel elasticity decreases with decreasing Mw, as expected from the results documented by Rochas et al. [14]. Such correlation is illustrated in Figure 4, which presents *G0* as a function of Mw for all carrageenans. The decrease in *G0* is less evident for the iota-carrageenan samples since the ultrasonication did not produce carrageenans with very different molecular masses. Overall, the results contradict the conclusions of the study of Souza et al. [15] performed with hybrid carrageenans dissolved in a different salt, NaCl. By comparing two highly sulphated hybrid carrageenans with similar chemical composition but different Mw, a larger *G0* was found for the gel formed with the polysaccharide having a smaller Mw [15]. Here, only two carrageenans offer such a comparison but, in KCl: the hybrid carrageenans from *G. pistillata* and the Gigartinale show significant amounts of **G4S-D2S** and **G4S-D2S,6S** and an equivalent range of initial Mw as in [15]. Such mismatch on the effect of Mw on *G0* is thus assigned to the difference in the type of salt used to induce gelation and, thus, to the cation specificity of the kappa-carrageenan blocks in the hybrids.

The data in Figure 4 allow for identifying a critical molecular mass in connection with the definition of Mc given by Rochas et al. [14]. The critical mass Mc below which no gel could be formed in the rheometer at 25 °C is here defined by the limits between the empty and the solid symbols in Figure 4. The values of Mc gathered in Table 1 for each carrageenan show that this critical mass depends on the chemical structure of carrageenans. This is the main result of this study. The large error bar associated with the Mc measured with the kappa-carrageenan hinders any clear conclusion on the difference between the two tested homopolymers. However, the data in Figure 4 suggest that kappa-carrageenan has a smaller Mc than iota-carrageenan, as the latter forms liquid at equivalent Mw, and this trend is confirmed by the low Mc found for the kappa-2-carrageenan and the less sulphated hybrid carrageenan, which both show larger contents of **G4S-DA**, the disaccharides making up the kappa-carrageenan. Thus, although iota-carrageenan needs fewer diads to adopt a helical conformation [20], it does need longer chains to network the helical aggregates. The value of Mc = 0.21 MDa found for the kappa-carrageenan is five times larger than the Mc = 0.04 MDa extrapolated by Rochas et al. [14] based on the assumption of critical overlap concentrations for chains in coil conformation. A possible explanation for the mismatch in Mc is that the kappa-carrageenan used here has 10 mol.% of **G4S-DA2S**, whereas the sample used in [14] would be a pure kappa-carrageenan. However, the authors did not give any detail on the chemical structure of their samples, though the same range of Mw is studied. Interestingly, the most sulphated carrageenans, from iota-carrageenan and from *G. pistillata*, show the largest Mc. This was expected for *G. pistillata* based on the concept of defects given in the introduction. The presence of **G4S-D2S,6S** and **G4S-D6S** limits the formation of helices [8], and longer chains are thus needed to network the smaller assemblies of helices. However, no defect is present in the chains of iota-carrageenan. But, without further knowledge on both the gel mechanism and the gel structure–elasticity relationships in carrageenans [10,11,12], it is hard to explain so far why carrageenans showing larger amounts of **G4S-DA** show systematically smaller Mc in Table 1.

The Mw independent elastic Young moduli *E* of gels produced by Rochas et al. correspond to shear elastic moduli of the order of *G0* ~ 13 kPa, assuming *E* = 3*G0* as in [14]. This is three times larger than the largest moduli found in Figure 4 for kappa-carrageenan gels. However, if the vertical axis in Figure 4 of [14] is in decimal logarithmic scale, then the data presented in Table 2 for the kappa-carrageenan gels are a good match. The semi-logarithmic plot in Figure 4 also pinpoints a second critical mass, which can be identified as the Mmax of Rochas et al. [14]. Indeed, *G0* reaches a plateau when Mw > Mmax. The values of Mmax estimated from the plot for kappa-carrageenan give Mmax = 0.2 MDa, which is a fair match to the 0.18 MDa reported in [14]. The data in Figure 4 also suggest that Mmax depends on the blockiness of the carrageenan copolymer structure. The homopolymers, iota- and kappa-carrageenan, show equivalent Mmax on the order of 0.2 to 0.26 MDa. The more heterogeneous hybrid carrageenans with long enough blocks of **G4S-DA** show a smaller Mmax of the order of 0.15 to 0.18 MDa. As for the highly sulphated hybrid carrageenan from *G. pistillata*, Mmax nearly matches Mc. However, when the data in Figure 4 are plotted on linear scales, the occurrence of a potential limiting molecular mass Mmax above which *G0* saturates is much less convincing; see inset to Figure 4 and Table 2. Clearly, more rheological data for carrageenans with Mw > 0.2 MDa would be needed to settle the existence of Mmax.

### 2.4. Chemical Structure of the Ultrasonicated Carrageenans

In a recent study, it has been suggested that ultrasonication leads to a desulphatation of kappa-carrageenan diads concomitantly with chain scission [17]. This outcome is at odds with the common practice in the carrageenan literature to lower the viscosity of carrageenan solutions by ultrasonication prior to chemical analysis by proton NMR [20,22]. However, the conclusions of this study were sufficient to trigger a qualitative chemical analysis of the carrageenans produced with the longest sonication time *t*. The spectra from the FTIR analyses are produced in Figure 5.

No change in the chemical structures of sonicated carrageenans from *Cottonii*, *M. stellatus* and the Gigartinale was found, since the typical bands assigned to [25] total sulphate (S=O vibration at 1240 cm^−1^), **DA** (C-O vibration at 930 cm^−1^), **G4S** (C-O-SO_3_ vibrations at 845 cm^−1^) and **DA2S** (C-O-SO_3_ vibrations at 805 cm^−1^) are identified for all samples in Figure 5a. The FTIR analysis thus validates the Mc identified in Table 1 and the effect of the carrageenan chemical structure on Mc. The picture emerging from the comparison of curves in Figure 5b is different, since a new band in the region 1120–1105 cm^−1^ shows up in the spectra of the sonicated carrageenans. This band shows up in a region assigned to the stretching vibrations of C-O bonds in sugars [26]. Such bonds may result from either the break-up of the glycosidic bonds [19] or, as claimed elsewhere, from the break-up of the S=O bonds of the **G4S** due to ultrasound-induced generation of free radicals [17]. The exact mechanism for such bond break-up was not detailed [17], but double bonds are known to involve more energy than single bonds. Thus, it remains unclear why such S=O bands appear at short ultrasonication times (4 h in spectra of Figure 5b) and not at times as long as 20 h (see Figure 5a). Clearly, additional chemical characterization of carrageenan as a function of the sonication time is needed to settle these points.

## 3. Conclusions

The ultrasonication of five carrageenans having different chemical structures successfully produced polysaccharides with different Mw. This study showed that Mw reduction was faster for the more sulphated carrageenan, namely the iota-carrageenan sample. The PDI saturates at longer ultrasonication times around 2 for all tested carrageenans. The rheological properties of 1 *w*/*v*% carrageenans in 0.1 M KCl indicate that the critical mass Mc below which no gel can be formed is a function of the carrageenan chemical structure. Mc is the smallest for the kappa-2-carrageenan, whereas iota-carrageenan and highly sulphated chains need to be much longer to start gelling. No Mmax was identified within the set of Mw tested here, which calls for additional studies with carrageenans having molecular masses larger than 0.2 MDa. No chemical changes were observed for the kappa-carrageenans, the kappa-2-carrageenan and the hybrid carrageenan with fewer sulphate groups. However, the chemical structure of the two most sulphated carrageenans which also showed the least resistance to ultrasonication was changed. As such, future studies should also focus on the detailed chemical structure of these two more sensitive carrageenans with a view to clarifying whether the larger Mc found for these two polysaccharides is biased by a chemical modification occurring simultaneously with chain scission.

## 4. Materials and Methods

### 4.1. Materials

Two algal species, *Mastocarpus stellatus* and *Gigartina pistillata*, were purchased from Algaplus (Ilhavo, Portugal) to produce hybrid carrageenans with more heterogeneous disaccharide structures. Two commercial seaweeds belonging to the Solieriaceae family were donated by Cargill Texturing Solutions—Hydrocolloids (Carentan, France) to produce nearly homopolymers from *Spinosum* and *Cottonii*, whereas a more hybrid carrageenan was extracted from a donated seaweed of the Gigartinaceae family. The dried seaweeds were processed as received. Potassium hydroxide and potassium chloride were purchased from Sigma-Aldrich Química SL (Sintra, Portugal), whereas ethanol (96 *v*/*v*%) was purchased from local retailers.

### 4.2. Carrageenan Extraction and Utrasonication

A protocol optimized in earlier studies was adapted. Briefly, 1.5 g dried seaweed was mixed with 50 mL of distilled water at 80 °C during 3 h. The mixture was then diluted with 100 mL hot distilled water and homogenized in a blender before further heating at 90 °C for an additional hour. To separate the solid algal residues from the carrageenan-rich solution, the mixture was centrifuged at 8000 rpm for 10 min, the supernatant filtered and poured in a plastic mold to be dried in an oven with air convection at 50 °C overnight. The resulting film was solubilized in 3 *w*/*v*% KOH (2 g of film in 100 mL) at 85 °C during 3 h to perform the alkali-induced conversion of more sulphated carrageenans into **G4S-DA** or **G4S-DA2S**. The solution was then precipitated in iced ethanol. The resulting precipitate was then filtered and dried overnight in an air convection oven at 50 °C. Triplicated extractions were performed to produce at least 6 g (dry weight) of hybrid carrageenans.

For the ultrasonication, carrageenan solutions were prepared in distilled water at a concentration of 0.5 *w*/*v*%. Though polymer concentration is usually not reported as a critical parameter for speeding up chain scission in polysaccharides [19], lower concentrations were found here to lead to a faster breakdown of carrageenan chains, in agreement with earlier reports on kappa-carrageenans [17]. A concentration of 0.5 *w*/*v*% was thus a good compromise between acceptable ultrasonication time and amount of processed material to be produced for subsequent characterization. Carrageenan dissolution was ensured by mixing at 80 °C for 1 h under vigorous stirring. The solutions were treated during varying time *t* in an ultrasonic bath (DCG-300H, MCR Ltd., Harlow, UK), operated at 40 kHz and an ultrasonic power of 400 W in pulsed mode and temperature-controlled at 60 °C. After time *t*, aliquots were taken from solutions for further carrageenan characterization, either dried into carrageenan films for NMR and Mw determination or directly as solutions for gel testing and rheometry.

### 4.3. Carrageenan Characterization and Gel Testing

Proton Nuclear Magnetic Resonance (NMR) spectroscopy at 70 °C using a Bruker Avance III spectrometer (Billerica, MA, USA) at 400 MHz was performed on carrageenan solutions (0.5 wt.% in D_2_O). Proton NMR peaks (at 5.10 ppm, 5.28 ppm, 5.5 ppm and 5.25 ppm, respectively, assigned to **G4S-DA**, **G4S-DA2S**, **G4S-D2S,6S** and **G4S-D2S** [20]) were used to compute the respective molar fractions (mol.%) of the carrageenan diads.

Size exclusion chromatography (SEC, Waters 600 apparatus, with a Waters 2410 differential refractive index detector, Waters, Lisboa, Portugal) was used to determine the molecular mass distribution of extracted and sonicated carrageenans. The SEC was coupled to a PolySep-GFC-P Linear column (Phenomenex, Alcobendas, Spain) and calibrated with pullulan (Shodex, Munich, Germany) ranging from 6300 to 642,000 g/mol. Carrageenan solutions (0.1 wt.% in 0.1 M NaCl) were injected into the column at 40 °C and Mw and PDI were measured in duplicate.

Aliquots sampled from the ultrasound bath were evaporated to reach a carrageenan concentration of 1 wt.%. KCl was then added to form a carrageenan solution at 0.1 M, which was stirred for 15 min at 80 °C and then allowed to cool at room temperature for 24 h to induce gel formation and mature a gel structure. As a control, a non-sonicated sample (*t* = 0 h) was included, which successfully formed gels for all carrageenans tested. Examples of aliquots after KCl addition are given in Appendix A. The gelling properties of the samples following ultrasonication were compared, revealing that some samples lost their ability to gel after 4 h of treatment, while others remained gelled until 20 h. The ultrasonication duration *t* of a carrageenan solution was stopped when an aliquot sample remained liquid after the gel test.

The aliquot sample that did not gel at the longest time *t* tested was dried to form a film, as detailed above, and the film was loaded into the Attenuated Transmission Reflectance accessory of a Fourier Transform InfraRed spectrometer (PerkinElmer Spectrum 100, Shelton, CT, USA). ATR-FTIR spectra were acquired by averaging 16 scans from 4000 to 600 cm^−1^ with a 1 cm^−1^ resolution. The assignment of FTIR bands was based on the bands listed in [25].

### 4.4. Rheological Characterization of Carrageenan Solutions and Gels

After the gelation test, the carrageenan samples subjected to different ultrasonication times *t*, as well as the control sample (*t* = 0 h), were characterized by rotational rheometry. Gel samples were loaded into the parallel plates (diameter of 40 mm) of a stress-controlled rheometer (MCR 302, Anton Paar, Graz, Austria) preheated to 85 °C. After sample loading and melting, the plate was lowered to the measuring position and dodecane was used to tap the geometry’s rim and prevent water loss. During the cooling performed down to 25 °C in 5500 s, a small amplitude (0.1%) oscillatory shear strain (SAOS) was applied at 1 Hz to follow the thermal evolution of the storage (*G′*) and loss (*G″*) moduli. In addition, the normal force was set to 0 N during cooling to allow for the volume change during the liquid-to-solid transition while keeping contact between gel and shearing plates, thus avoiding possible artefacts in the measurements of G′ and G″ [21]. From the thermal dependence of the gap, the temperature for onset of gelation, *Ton*, is measured as the temperature where the gap stops following a linear thermal expansion due to the shrinkage of the sample upon gel formation. After cooling, a frequency (logarithmic) sweep was performed, from 100 Hz down to 0.02 Hz with a 0.1% strain amplitude, to measure the gels’ mechanical spectra and determine the gels’ linear shear elastic modulus *G0* as the value of G′ measured at 1 Hz. For liquid samples, another protocol was followed. After loading at 25 °C, samples were left to equilibrate during 5 min, while G′ and G″ were measured at 1 Hz with an SAOS of 10%. Then, a frequency sweep was performed as above but with an amplitude of 50% to generate large enough torque for the measurement of G′ and G″, from which G′ obtained at 1 Hz was reported as a measure of the viscoelasticity *G0* of the solutions. *G0* showed a satisfactory match with the values of G′ measured during equilibration with a strain of 10%, thus ensuring that *G0* is obtained from the linear regime of viscoelasticity.

## Figures and Tables

**Figure 1 gels-11-00077-f001:**
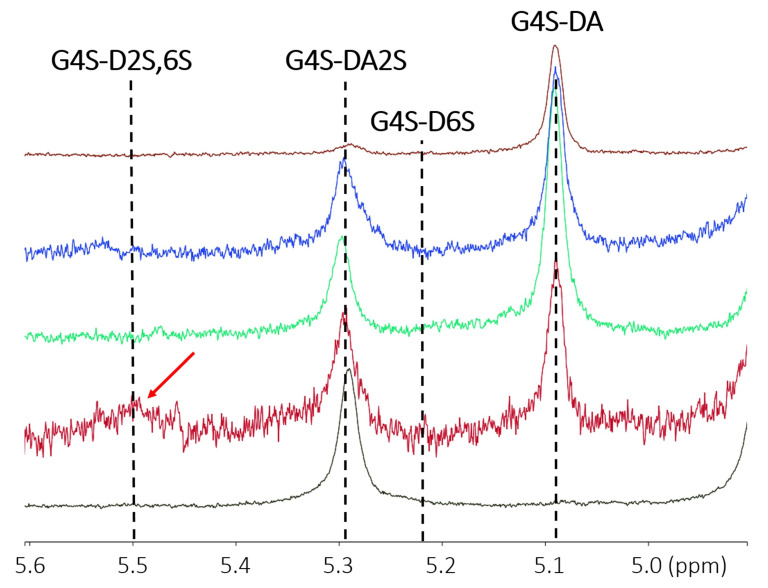
Proton NMR spectra of extracted and alkali-modified carrageenans. From bottom to top: *Spinosum*, *G. pistillata*, *M. stellatus*, *Gigartinale* and *Cottonii*. The red arrow indicates the peak assigned to nu-carrageenan in the polysaccharide from *G. pistillata*. Full spectra are available in Appendix A.

**Figure 2 gels-11-00077-f002:**
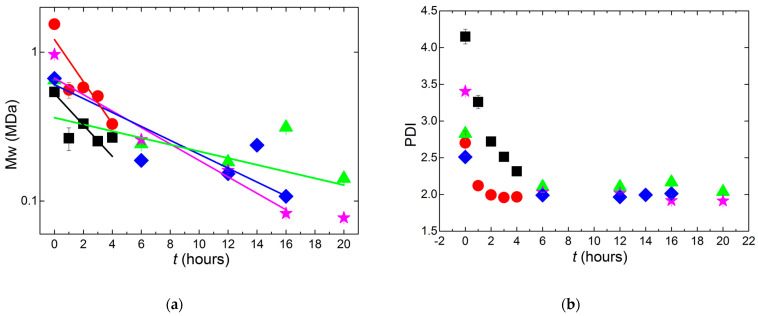
Kinetics of chain scission of carrageenans submitted to ultrasonication during a time *t*: iota-carrageenan from *Spinosium* (squares), highly sulphated hybrid carrageenan from *G. pistillata* (circles), kappa-2-carrageenan from *M. stellatus* (triangles), hybrid carrageenan from a Gigartinale (diamonds) and kappa-carrageenan from *Cottonii* (stars). (**a**) Time dependence of the molecular mass Mw. (**b**) Time dependence of the polydispersity index, PDI, computed from the ratio Mw/Mn, where Mn is the number average of the molecular mass distribution. The lines in (**a**) are exponential fits to the data (see text).

**Figure 3 gels-11-00077-f003:**
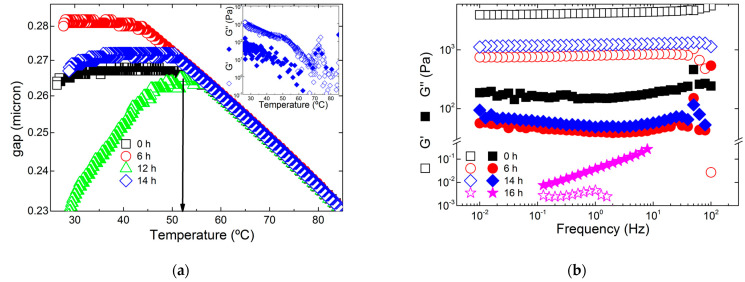
(**a**) Temperature dependence of the rheometer’s gap during the cooling of hot solutions of hybrid carrageenans from the Gigartinale and ultrasonicated during the indicated times. The arrow points to the temperature *Ton* defining the onset of gelation. The inset shows the evolution of G′ and G″ measured at 1 Hz during cooling. (**b**) Mechanical spectra of the gelled carrageenans from the Gigartinale and the liquid carrageenan obtained after 16 h of ultrasonication.

**Figure 4 gels-11-00077-f004:**
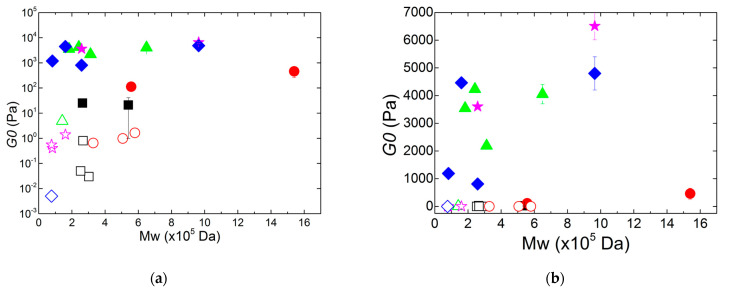
Shear elastic modulus *G0* of gels (solid symbols) and solutions (empty symbols) of carrageenans produced with the corresponding Mw obtained by ultrasonication: iota-carrageenan from *Spinosium* (squares), highly sulphated hybrid carrageenan from *G. pistillata* (circles), kappa-2-carrageenan from *M. stellatus* (triangles), hybrid carrageenan from a Gigartinale (diamonds) and kappa-carrageenan from *Cottonii* (stars). (**a**) Semi-logarithmic plot. (**b**) Rescaling of the data plotted in (**a**) in linear axes.

**Figure 5 gels-11-00077-f005:**
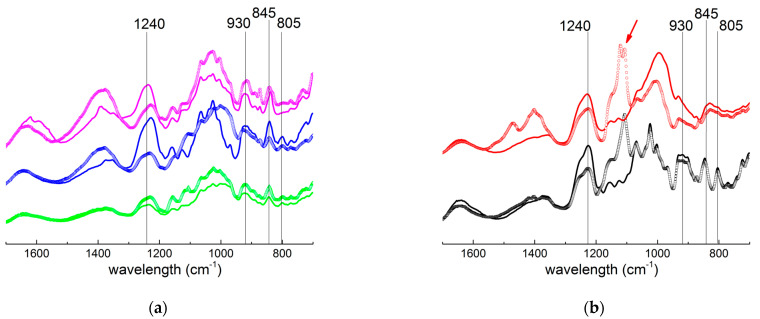
FTIR spectra of virgin (lines) and ultrasonicated (symbols) carrageenans from (**a**) bottom to top: *M. stellatus*, Gigartinale and kappa-carrageenan and from (**b**) bottom to top: iota-carrageenan and *G. pistillata*. Spectra are vertically shifted to facilitate a comparison. Vertical lines indicate the characteristic FTIR bands assigned to specific carrageenan diads. The arrow in (**b**) indicates the bands at 1120–1105 cm^−1^, which show up in sonicated samples. Full FTIR spectra are available in Appendix A.

**Table 1 gels-11-00077-t001:** Chemical compositions (in mol.%) of the hybrid carrageenans extracted from commercial seaweeds and critical mass Mc for 1 *w*/*v*% carrageenan in 0.1 M KCl.

Algae	G4S-DA	G4S-DA2S	G4S-D6S	G4S-D2S,6S	Mc (10^5^ Da)
*Spinosum*	4.0 ± 2.5	96 ± 5	0	0	2.8 ± 0.2
*Gigartina pistillata*	42 ± 1.8	39.0 ± 2.1	0	19 ± 5	5.7 ± 0.1
*Mastocarpus stellatus*	69.0 ± 0.4	31.0 ± 2.2	0	0	1.6 ± 0.2
*Gigartinale*	56.0 ± 2.9	34.0 ± 4.7	4 ± 1	6 ± 1	1.3 ± 0.2
*Cottonii*	90.0 ± 4.5	10 ± 5	0	0	2.1 ± 0.5

**Table 2 gels-11-00077-t002:** Rheological parameters inferred from the cooling of hot carrageenan solutions (*Ton*) and from the mechanical spectra (*G0*) measured at 1 Hz and 25 °C, with the indicated error bars, computed from duplicate experiments performed with non-sonicated samples of the order of the 10% accepted for rheometry [24]. Entries with no data indicate a liquid sample.

Carrageenans		*Ton* (°C)	*G0* (Pa)
Algae	Time (h)		
*Spinosum*	0	35 ± 2	21 ± 20
	1	30 ± 1	25
	2	-	0.03
	3	-	0.05
	4	-	0.8
*G. pistillata*	0	52 ± 1	460 ± 200
	1	<25	113
	2	-	1.64
	3	-	0.98
	4	-	0.65
*M. stellatus*	0	54 ± 1	4050 ± 350
	6	55 ± 1	4232
	12	52 ± 1	3543
	16	50 ± 0.5	2189
	20	-	4.85
Gigartinale	0	56 ± 1.5	4800 ± 600
	6	43 ± 1	810
	12	50 ± 0.5	4461
	14	56 ± 1	1188
	16	-	0.005
*Cottonii*	0	57 ± 1	6000 ± 500
	6	51.5 ± 1.5	3604
	12	-	1.4
	16	-	0.4
	20	-	0.55

## Data Availability

The original contributions presented in this study are included in the article/Appendix A. Further inquiries can be directed to the corresponding author.

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
