# Peer review of "Role of the Molecular Mass on the Elastic Properties of Hybrid Carrageenan Hydrogels"

_gels, 2025, doi:10.3390/gels11010077_

Round 1

Reviewer 1 Report

Comments and Suggestions for Authors

This research article is devoted to the study of biopolymers based on various carrageenan derivatives, as well as the influence of the molecular weight of the gels obtained on the elastic properties of these compounds. The authors compare carrageenans modified in various ways and studied the kinetics of degradation of the obtained gels and the effect of ultrasound exposure on the gelation process. A number of patterns have been established, which will be used by the authors as the basis for their further research in the field of biopolymers, as the authors say in the conclusion of their study.

Despite this, there are a number of issues that are recommended for correction:

1.      A large number of the mentioned literary sources were published more than 5 years ago, and therefore it is recommended (if possible) to update the literary data to more recent ones.

2.      Figure 1 and the following Table 1 and their discussion indicate the presence of a large number of G4S-D2S,6S, capable of interrupting the formation of the spiral conformation of G4S-DA and G4S-DA2S. At the same time, there is no peak in the figure itself, signaling the presence of G4S-D2S,6S for Gigartina pistillata, which casts doubt on the plausibility of further research. In this regard, it is recommended to clarify the results of this experiment and identify the existing error, or provide an explanation to this figure with the designation of the peak, which is discussed in the further discussion of the results.

3.      Figure 2a shows a graph of the dependence of the decrease in the molecular weight of the studied samples on the time of ultrasonic exposure. A trend line is given for the sample, kappa-carrageenan, obtained from Cottonii. What is the reason for its presence only for this sample under study, and why are similar trend lines not constructed for the 4 remaining samples?

4.      Figure 3a shows the evolution of the coefficients G’ and G’ in an additional box. This insert is difficult to read due to the selected markers, so it is recommended to change the color scheme for the coefficients to make the image more contrasting and readable, or to abandon the use of large markers in this insert to improve image quality.

5.      Figure 4 – is it necessary to combine two graphs in a similar way, or is it possible to divide them into 2 separate graphs, indicated by sub-items a) and b).

6.      Despite the mention of the scientific significance of the study, the practical applicability of such gels is not indicated, which also underlines the importance of research in this field. I would also like to know about the possibility of biodegradation of such gels, at least in the form of information provided in the introductory part of the article, if this does not affect the integrity of the work.

As a result, this work is of great importance in the study of carrageenan-based biopolymer structures, but nevertheless has a number of significant drawbacks and requires significant improvements. This work is recommended for publication after corrections have been made.

Author Response

see attached document

Reviewer 2 Report

Comments and Suggestions for Authors

The manuscript entitled “Role of the molecular mass on the elastic properties of hybrid carrageenan hydrogels

The study examines the effects of molecular weight reduction on carrageenans through ultrasonication, finding slower chain scission for those richer in kappa-carrageenan units. It identifies a critical molecular mass (Mc) below which gel formation is not possible, with Mc being lower for kappa- and kappa-2 carrageenans (0.13-0.21 MDa) and higher for iota- (0.280 MDa) and hybrid carrageenans (0.57 MDa). No plateau in gel elasticity with respect to molecular weight is observed, indicating continuous changes in gel properties.

The manuscript is overall well-written but would benefit from some minor revisions (outlined below). The topic of the study is highly engaging and fits well within the journal's scope. The experimental design was thorough, and the execution was sound, leading to conclusions that are well-supported. I recommend accepting the paper for publication once the following points have been addressed:

1.     For the NMR and FTIR spectra (Figures 1 and 5), the authors should include the full spectrum for all the samples in the Supporting Information, as this would help assess the purity of the compounds.

2.     In the abstract, the authors should provide the full names of K+ and KCl before using their abbreviations.

3.     The authors should increase the thickness of the peaks and the boldness of the figure titles to enhance the quality of the figures.

4.     The authors should include images of the solution after ultrasonication in the Supporting Information.

5.     The authors should include the DOI in references 1 and 16.

6.     The authors should review the referencing format, as in some instances the year is bolded while in others it is not. The formatting should be consistent throughout.

Author Response

see attached document

Round 2

Reviewer 1 Report

Comments and Suggestions for Authors

The authors have corrected all the comments. The article can be published.